# Epidemiology of *Plasmodium malariae* and *Plasmodium ovale* spp. in Kinshasa Province, Democratic Republic of Congo

Rachel Sendor [1] ✉, Kristin Banek [2], Melchior M. Kashamuka[3], Nono Mvuama [3], Joseph A. Bala[3], Marthe Nkalani[3], Georges Kihuma[3], Joseph Atibu[3], Kyaw L. Thwai[2], W. Matthew Svec[4], Varun Goel[5,6], Tommy Nseka[3], Jessica T. Lin[2,7], Jeffrey A. Bailey [8], Michael Emch [1,5], Margaret Carrel[9], Jonathan J. Juliano[1,2,7], Antoinette Tshefu[3,10] & Jonathan B. Parr [2,7,10] ✉

Reports suggest non-falciparum species are an underappreciated cause of malaria in sub-Saharan Africa but their epidemiology is ill-defined, particularly in highly malaria-endemic regions. We estimated incidence and prevalence of PCR-confirmed non-falciparum and *Plasmodium falciparum* malaria infections within a longitudinal study conducted in Kinshasa, Democratic Republic of Congo (DRC) between 2015-2017. Children and adults were sampled at biannual household surveys and routine clinic visits. Among 9,089 samples from 1,565 participants, incidences of *P. malariae, P. ovale* spp., and *P. falciparum* infections by 1-year were 7.8% (95% CI: 6.4%-9.1%), 4.8% (95% CI: 3.7%-5.9%) and 57.5% (95% CI: 54.4%-60.5%), respectively. Non-falciparum prevalences were higher in school-age children, rural and peri-urban sites, and *P. falciparum* co-infections. *P. falciparum* remains the primary driver of malaria in the DRC, though non-falciparum species also pose an infection risk. As *P. falciparum* interventions gain traction in high-burden settings, continued surveillance and improved understanding of non-falciparum infections are warranted.

Reports of non-falciparum malaria caused by *Plasmodium malariae* and *Plasmodium ovale* spp. have increased across sub-Saharan Africa, where more than 95% of global malaria cases and deaths occur[1]. *Plasmodium falciparum* is the primary cause of malaria morbidity and mortality in the region. However, molecular surveys have confirmed co-circulating non-falciparum species, and suggest a rise in prevalence in regions where *P. falciparum* has declined[2–6]. Despite this, our understanding of infection risk and clinical burden posed by non-falciparum species is limited, particularly within regions of high *P. falciparum* transmission.

The Democratic Republic of Congo (DRC) is one of 11 malaria "High Burden, High Impact" countries designated by the World Health Organization (WHO)[7], indicating a critical need to understand the full landscape of malaria transmission and infection in the country. The

[1]Department of Epidemiology, Gillings School of Global Public Health, University of North Carolina at Chapel Hill, Chapel Hill, NC, USA. [2]Institute for Global Health and Infectious Diseases, University of North Carolina at Chapel Hill, Chapel Hill, NC, USA. [3]Ecole de Santé Publique, Faculté de Médecine, University of Kinshasa, Kinshasa, Democratic Republic of the Congo. [4]University of North Carolina at Chapel Hill, Chapel Hill, NC, USA. [5]Department of Geography, University of North Carolina at Chapel Hill, Chapel Hill, NC, USA. [6]Carolina Population Center, University of North Carolina at Chapel Hill, Chapel Hill, NC, USA. [7]Division of Infectious Diseases, School of Medicine, University of North Carolina at Chapel Hill, Chapel Hill, NC, USA. [8]Department of Pathology and Laboratory Medicine and Center for Computational Molecular Biology, Brown University, Providence, RI, USA. [9]Department of Geographical and Sustainability Sciences, University of Iowa, Iowa City, IA, USA. [10]These authors jointly supervised this work: Antoinette Tshefu, Jonathan B. Parr. ✉e-mail: rachel.sendor@unc.edu; jonathan_parr@med.unc.edu

DRC harbors the second highest burden of malaria worldwide, accounting for 12% of global malaria cases and 13% of malaria-related deaths as of 2020[1]. Transmission is largely stable and perennial throughout the country, with an estimated 97% of the population living in high malaria transmission regions for eight or more months per year[8–10]. While the majority of the country is characterized as hyper- and holo-endemic for malaria[8,9,11], transmission can vary within and across provinces due to environmental and geographical factors such as urbanicity, population density, land use, temperature, and rainfall[10,12]. In Kinshasa Province, home to the capital city of Kinshasa, malaria transmission is lower within the central urban areas, and increases in the outer peri-urban and rural zones[10].

While *P. falciparum* accounts for the vast majority of infections in the DRC, *P. malariae* and *P. ovale* spp. have been documented at low-level prevalences[8,13–15]; however, their epidemiology and clinical impact remains poorly understood. Existing non-falciparum prevalence estimates were largely derived from cross-sectional studies, limiting risk assessment, or focused on a singular age group or symptomatic status, which restricts broader generalizability[13,15–21]. The absence of reliable field diagnostics for these less-common infections complicates surveillance efforts and case management. Gold-standard microscopy methods can detect non-falciparum parasites, but insufficient training and specialization ultimately limits the sensitivity and specificity for non-falciparum infections[22]. Widely-used malaria rapid diagnostic tests (RDTs) in Africa cannot distinguish *P. malariae* or *P. ovale* spp. infections. While most RDTs detect *P. falciparum*-specific histidine-rich protein 2 (HRP2), and some also detect *Plasmodium* lactate dehydrogenase (LDH), no *P. malariae* and *P. ovale* spp.-specific RDTs are currently available. Further, clinical diagnosis of a non-falciparum infection is not feasible in the absence of symptoms attributed specifically to *P. malariae* or *P. ovale* spp. These infections are complicated to detect in malaria-endemic regions such as the DRC where *P. falciparum* co-infection is common, making it difficult to distinguish symptoms caused specifically by these non-falciparum species. *P. malariae* infections have been associated with acute febrile illness and anemia[23–25] as well as chronic nephrotic syndrome[26,27]. *P. ovale* spp. infections commonly result in low morbidity, although severe complications have been documented in case reports[28,29]. The clinical relevance of *P. falciparum* co-infection with non-falciparum species is also unclear, with posited suppressive effects observed in one study[4], while another detected severe malaria among mixed-infections[24]. These diagnostic challenges are exacerbated by low parasite densities characteristic of non-falciparum infections, which require the use of sensitive laboratory assays that are not suitable for routine field use.

*P. malariae* and *P. ovale* spp. infections have occasionally been dismissed as rare infections that are not clinically impactful in high-burden countries like the DRC. However, improved characterization of their epidemiology and clinical implications is needed to determine their proper place in malaria programmatic and surveillance efforts. Leveraging a 34-month longitudinal cohort study conducted across diverse sites in Kinshasa Province, DRC, we sought to determine the epidemiology of *P. malariae* and *P. ovale* spp. infections in the context of high *P. falciparum* transmission. We performed high-throughput, real-time polymerase chain reaction (PCR) on samples collected from both asymptomatic and symptomatic participants of all ages as part of a large longitudinal study of non-falciparum malaria infection. We estimated incidence, associated factors, and clinical features associated with *P. malariae* and *P. ovale* spp. infections over time, and as compared to *P. falciparum* infections.

## Results
### Study population
A total of 1591 individuals were enrolled in the parent cohort across 242 households and seven sites in Kinshasa Province, DRC. Among these, 1565 (98.4%) participants had a baseline dried blood spot (DBS) sample

available for analysis and were included in this study, contributing 5682 total survey visits. Participant follow-up by analysis population is displayed in Fig. 1. In the main survey population, 76% of participants completed all three study follow-ups; loss-to-follow-up differed by regional health area, with a higher proportion observed in peri-urban (28%, $n = 143$) and urban (30%, $n = 116$) health areas compared to rural (18%, $n = 116$). Sixty-seven percent ($n = 1050$) of those in the main survey population also had ≥1 symptomatic clinic visit during the 34-month passive follow-up period and were included in the clinic-based analysis, comprising 218 (90.1%) of the 242 enrolled households. Participants in the clinic subpopulation had 3407 total clinic visits, with a median (interquartile range [IQR]; min-max) of 2 (1–4; 1–19) visits per person and 11 (5–22; 1–74) per household.

Baseline demographic and clinical characteristics for the survey population are summarized in Table 1. The median (IQR) age of participants at baseline was 14 (6–31) years, with 19% of participants aged <5 years, and 32% school-aged (between 5 and 14 years). The majority of participants in the survey population were female (55%), and 42% were living in rural sites. Approximately 73% of the population lived in a household with at least one bed net, although only 45% reported sleeping under a bed net. Twenty-four percent of participants self-reported having a fever in the week prior to baseline, 25% reported experiencing malaria symptoms in the prior 6 months and taking antimalarials, and 27% of participants had a positive malaria RDT at the baseline survey.

Baseline characteristics were similar between those also included in the symptomatic clinic subpopulation, and the full survey population (Supplementary Table 1). The median (IQR) age of participants in the clinic subpopulation was 12 (5–30) years, with 22% aged <5 years, and 34% school-aged (5–14 years). There was also a higher proportion of women (56%) than men in the clinic subpopulation, and higher proportions of participants living in rural (22%) or peri-urban (32%) sites compared to urban sites (46%), as in the broader survey population (Supplementary Table 2).

Participant characteristics across follow-up visits are summarized in Supplementary Table 3 and Supplementary Table 4. Overall, 97.0% ($n = 1518$) and 99.6% ($n = 1559$) of participants were PCR-negative for a *P. malariae* and *P. ovale* spp. infection at baseline, respectively; 69.1% ($n = 1081$) were PCR-negative for a *P. falciparum* infection at baseline. These PCR-negative participants comprised at-risk populations for malaria incidence estimation.

### Incidence of non-falciparum malaria infections
The estimated 1-year cumulative incidences of *P. malariae* and *P. ovale* spp. infections were 7.8% (95% CI: 6.4%–9.1%) and 4.8% (95% CI: 3.7%–5.9%), respectively, encompassing the first incident infections detected at survey or clinic visits. Comparatively, 57.5% (95% CI: 54.4%–60.5%) of at-risk participants acquired a *P. falciparum* infection by 1-year (Fig. 2). Risk of *P. malariae* and *P. ovale* spp. was similar within the first year of follow-up, but appears to increase at, and following, the second study follow-up visit around 1-year from baseline, after which the risk of a *P. malariae* infection surpassed that of *P. ovale* spp. Time to the first detected infection was faster for *P. falciparum* than *P. malariae* or *P. ovale* infections throughout the whole study period, with at least half of the total at-risk population experiencing a *P. falciparum* malaria infection by 12 months from baseline. Step-wise increases in incidence depicted in Fig. 2 represent infections detected at periodic household surveys; symptomatic infections detected through passive, clinic-based surveillance filled in gaps between surveys, as indicated by steady inclines between step-ups. The 1-year cumulative incidences of first *P. malariae* and *P. ovale* spp. infections detected only at active surveillance household surveys were approximately 4.9% (95% CI: 3.7%–6.0%) and 3.4% (95% CI: 2.4%–4.3%), respectively, including single- and mixed-species infections, whereas the cumulative

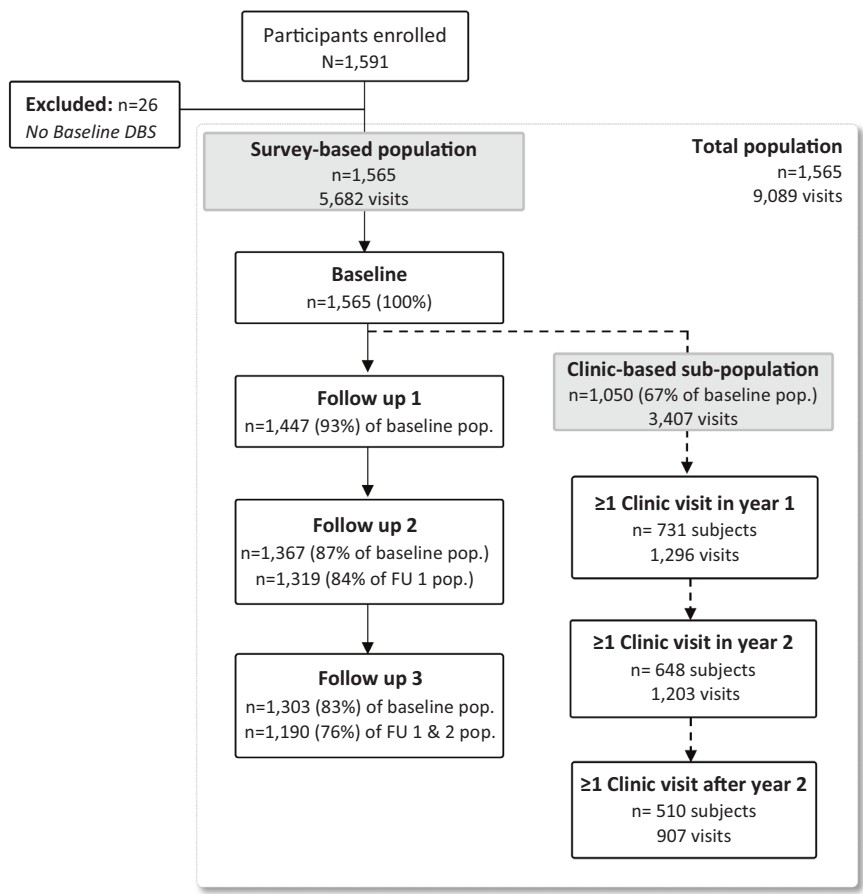

**Fig. 1 | Study population.** The study population comprised 1565 enrolled participants who had a DBS sample at the baseline visit. Participants without a baseline DBS were excluded from analyses. Participant loss-to-follow-up among the survey-based population is shown, with 76% completing all three follow-up visits. Sixty-seven percent of participants in the survey population visited a study health clinic at least once during follow-up, comprising the clinic-based subpopulation. Study health clinic visit frequency across the full study period is shown. The total population encompassed all survey and clinic-based study visits. DBS dried blood spot, FU follow-up.

incidence of a *P. falciparum* infection by 1-year was 42.8% (95% CI: 39.6%–45.8%) within the survey population (Table 2).

Cumulative incidences of first *P. malariae, P. ovale* spp., and *P. falciparum* infections stratified by age and sex are depicted in Fig. 3. Within the total study population, cumulative incidences of first *P. malariae* and *P. ovale* spp. infection were higher among children <5 and school-aged children 5–14 years old compared to participants aged ≥15 years (*Pm*: $p < 0.001$; *Po*: $p < 0.001$). Time to the first detected *P. ovale* spp. infection was similar between children <5 and 5–14 years throughout follow-up, whereas school-aged children 5–14 years old experienced a faster time to *P. malariae* infection, particularly towards the end of follow-up driven by symptomatic clinic-based infections. The cumulative incidence of first *P. falciparum* infection was also higher among children <5 and 5–14 years old, compared to adults aged ≥15, although *P. falciparum* incidences were high (≥50%) across all age categories by 1-year ($p < 0.001$). No differences in the cumulative incidence of *P. ovale* spp. and *P. falciparum* infection by sex were detected ($p = 0.73$ and $p = 0.62$, respectively), although *P. malariae* infection incidence was higher among males than females ($p = 0.03$).

**Prevalence of non-falciparum and *P. falciparum* infections**
Among 1565 participants across all baseline and follow-up household surveys in the approximately 2-year active surveillance period, 159 (10.2%) participants had 186 PCR-confirmed *P. malariae* infections, 70 (4.5%) participants had 78 *P. ovale* spp. infections, and 990 (63.3%) participants had 1976 *P. falciparum* infections detected. Of all 5682 tested samples collected from these participants across household follow-up surveys, 3.3% (95% CI: 2.8%–3.8%) were PCR-positive for a *P. malariae* infection, 1.4% (95% CI: 1.1%–1.7%) for a *P. ovale* spp. infection, and 34.9% (95% CI: 33.2%–36.5%) for a *P. falciparum* infection. The majority of *P. malariae* and *P. ovale* spp. infections in the survey-based population were mixed-species (*P. malariae*: 73.7% [$n = 137/186$]; *P. ovale* spp.: 78.2% [$n = 61/78$]), and were predominately co-infected with *P. falciparum*. In contrast, the vast majority of *P. falciparum* infections were single-species (90.5%; $n = 1788/1976$).

*P. malariae* prevalence across household survey visits remained steady throughout active follow-up, fluctuating between 3.0% at baseline, 2.4% at follow-up 1, 4.1% at follow-up 2, and 3.7% at the final household survey ($p = 0.06$; Supplementary Table 3). Similarly, *P. ovale* spp. prevalence remained relatively low across follow-up, with 0.4% of the population infected with *P. ovale* spp. at baseline, increasing to 1.9% and 2.0% at follow-ups 1 and 2, and ending at a 1.4% prevalence by the final follow-up ($p < 0.01$; Supplementary Table 3). *P. falciparum* prevalence increased slightly over time, with the highest prevalence (39.6%) detected at the second follow-up visit in the rainy season ($p < 0.01$; Supplementary Table 3).

Similar prevalences to those detected in the survey population were observed among the symptomatic clinic subpopulation, despite longer clinic-based follow-up. Among 1050 participants who visited study clinics in the approximately 3-year clinic-based surveillance follow-up period, a total of 120 (11.4%) patients had 135 *P. malariae* infections detected, and 80 (7.6%) patients had 95 *P. ovale* spp. infections detected. However, prevalence of *P. falciparum* was higher in symptomatic clinic patients compared to the survey population, with

**Table 1 | Baseline participant characteristics by species –survey population**

| Baseline participant characteristics no. (%) | Total survey population N = 1565 | Baseline malaria infection by species | | |
|---|---|---|---|---|
| | | *P. malariae* PCR pos. n = 47 | *P. ovale* spp. PCR pos. n = 6 | *P. falciparum* PCR pos. n = 484 |
| **Age (years)** | | | | |
| Median (IQR) | 14 (6–31) | 9 (6–13) | 13 (5–22) | 12 (7–18) |
| <5 | 302 (19.3) | 6 (12.8) | 1 (16.7) | 64 (13.2) |
| 5–14 | 500 (31.9) | 30 (63.8) | 2 (33.3) | 240 (49.6) |
| 15+ | 763 (48.8) | 11 (23.4) | 3 (50.0) | 180 (37.2) |
| **Sex** | | | | |
| Female | 863 (55.1) | 23 (48.9) | 5 (83.3) | 245 (50.6) |
| Male | 702 (44.9) | 24 (51.1) | 1 (16.7) | 239 (49.4) |
| **Rurality of household[a]** | | | | |
| Rural | 663 (42.4) | 28 (59.6) | 6 (100) | 279 (57.6) |
| Peri-urban | 517 (33.0) | 17 (36.2) | 0 (0.0) | 190 (39.3) |
| Urban | 385 (24.6) | 2 (4.3) | 0 (0.0) | 15 (3.1) |
| Fever in prior week | 382 (24.4) | 11 (24.4) | 4 (66.7) | 151 (31.2) |
| RDT+ | 429 (27.4) | 27 (60.0) | 3 (50.0) | 363 (75.0) |
| Household owns a bed net | 1135 (72.5) | 32 (68.1) | 4 (66.7) | 347 (71.7) |
| Bed net use | 705 (45.0) | 14 (29.8) | 2 (33.3) | 209 (43.2) |
| Malaria sx. ≤6 mon. | 388 (24.8) | 7 (15.6) | 3 (50.0) | 117 (24.2) |
| Antimalarial use ≤6 mon. | 394 (25.2) | 8 (18.2) | 1 (16.7) | 102 (21.1) |
| **Wealth category** | | | | |
| Poorer/poorest | 632 (40.4) | 26 (55.3) | 2 (33.3) | 264 (54.5) |
| Average | 311 (19.9) | 10 (21.3) | 2 (33.3) | 107 (22.1) |
| Wealthier/ wealthiest | 622 (39.7) | 11 (23.4) | 2 (33.3) | 113 (23.3) |

*IQR* interquartile range, *PCR* polymerase chain reaction, *RDT* rapid diagnostic test, *Sx* symptoms.
[a]Bu health area is classified as rural, Kimpoko health area as peri-urban, and Voix du Peuple as urban.

789 (75.1%) patients having 2009 *P. falciparum* infections. From the clinic subpopulation, 4.0% (95% CI: 3.3%–4.7%) of all 3407 tested clinic samples across the clinic-based surveillance follow-up period were PCR-positive for a *P. malariae* infection, 2.8% (95% CI: 2.2%–3.4%) for a *P. ovale* spp. infection, and 58.7% (95% CI: 56.5%–60.8%) for a *P. falciparum* infection.

The timeline of all *P. malariae* and *P. ovale* spp. infections by participant throughout the full study is depicted in Supplementary Fig. 1. Multiple non-falciparum infections of the same species were detected over time at survey or clinic-based surveillance visits in 23.5% (n = 58/247) of all participants who had a *P. malariae* infection, and in 21.6% (n = 30/139) of those who had a *P. ovale* spp. infection during the 34-month study period. Three participants each had a *P. malariae* infection detected 4 times throughout follow-up. In comparison, *P. falciparum* infections re-occurred frequently, with 75.2% (n = 914/1216) of those who had a *P. falciparum* infection having more than one during follow-up, and 22.4% (n = 272/1216) having ≥5 *P. falciparum* infections detected during the 34-month study period.

Children had non-falciparum infections more frequently than adults. Prevalence of *P. malariae* infection in the survey-based population was highest among school-aged children aged 5–14 years at 5.6% (95% CI: 4.4%–6.7%), though still low overall, followed by children <5 years (3.4% [95% CI: 2.1%–4.7%]), and lowest among those aged ≥15 (1.7% [95% CI: 1.2%–2.2%]). Conversely, children <5 years experienced a slightly higher prevalence of *P. ovale* spp. infection compared to school-aged children (2.3% [95% CI: 1.3%–3.3%] vs. 1.9% [95% CI: 1.2%–2.7%]), although prevalences across both age groups were low

overall, and lower than *P. malariae* prevalence in these age groups; 0.7% (95% CI: 0.4%–1.0%) of adults ≥15 years had a *P. ovale* spp. infection at study survey visits.

Factors associated with the prevalence of *P. malariae* and *P. ovale* spp. infections are shown in Fig. 4 (and Supplementary Tables 8 and 9), and contrasted with *P. falciparum* prevalence differences (PD) in Supplementary Fig. 2. Within the survey-based population, *P. falciparum* coinfection was associated with an increased prevalence of both *P. malariae* (PD 0.05 [95% CI: 0.04–0.06]) and *P. ovale* spp. (PD: 0.02 [95% CI: 0.02–0.03]) infections. In comparison to those with average wealth, higher wealth was associated with a decreased prevalence of *P. malariae* (PD: -0.01 [95% CI: -0.03 to -0.002]) and *P. ovale* spp. (PD: -0.01 [95% CI: -0.02 to -0.00]) infection. School-aged children 5–14 years old were associated with a higher prevalence of *P. malariae* (PD: 0.04 [95% CI: 0.03–0.05]) and *P. ovale* spp. (PD: 0.01 [95% CI: 0.005–0.02]), infection as compared to participants 15 and older and also as compared to children aged <5 for *P. malariae* (PD: 0.02 [95% CI: 0.005–0.04]). Additionally, both *P. malariae* and *P. ovale* spp. infection prevalences were higher among participants living in rural (*Pm:* PD: 0.04 [95% CI: 0.03–0.05]; *Po:* PD: 0.02 [95% CI: 0.01–0.02]) and peri-urban (*Pm:* PD: 0.04 [95% CI: 0.03–0.05]; *Po:* PD: 0.02 [95% CI: 0.01–0.02]) sites as compared to an urban setting.

Similarly, in the clinic-based analysis, school-aged children were also associated with an increased prevalence of *P. malariae* (PD: 0.02 [95% CI: 0.01–0.04]) and *P. ovale* spp. (PD: 0.02; [95% CI: 0.01– 0.04]), infections, compared to adults ≥15 and older; however, no differences in prevalence of *P. malariae* and *P. ovale* spp. were observed between children <5 years and adults ≥15 and older in this symptomatic population (*Pm*: PD: 0.01 [95% CI: -0.01–0.02]; *Po*: PD: -0.001 [95% CI: -0.01–0.01]). In contrast to the survey population, no associations were observed with *P. falciparum* co-infection for either non-falciparum species in the clinic-based analysis (*Pm*: PD: 0.01 [95% CI: -0.01–0.02]; *Po*: PD: -0.01 [95% CI: -0.02–0.003]). *P. malariae* infection prevalences were higher among rural (PD: 0.05 [95% CI: 0.04–0.06]) and peri-urban (PD: 0.03 [95% CI: 0.02–0.05]) sites as compared to urban in the clinic-based analysis, whereas no difference in *P. ovale* spp. prevalence was observed between rural and urban settings among symptomatic participants (PD: 0.01 [95% CI: -0.002–0.02]).

Prevalence of self-reported fever in the week prior to the survey visit was similar for *P. malariae* (PD: -0.006 [95% CI:-0.018–0.006]) and *P. ovale* spp. infections (PD: 0.007 [95% CI: -0.002–0.017]), as compared to no species-specific malaria infection among the survey population (Fig. 4). In the symptomatic clinic subpopulation, concurrent anemia of any severity was associated with *P. malariae* infection prevalence (PD: 0.015 [95% CI: 0.001–0.029]), although associations were no longer significant when stratifying by anemia severity, comparing moderate-to-severe cases vs. mild or no anemia for *P. malariae* (PD: 0.012 [95% CI: -0.006–0.029]), or for *P. ovale* spp. (PD: 0.002 [95% CI:-0.013–0.018].

**Differences in parasite densities by species mixed infections**
Estimated median (IQR) parasite densities were low (<50 parasites [p]/μL) among all non-falciparum infections in the study (*P. malariae*: 25.7 [7.7–119] p/μL, n = 307; *P. ovale*: 10.2 [2.7–47.4] p/μL, n = 164), while median (IQR) *P. falciparum* parasite densities in the total population were higher at 267 p/μL (18.8–4526 p/μL, n = 3730). Non-falciparum parasite densities remained low when stratified by survey population (*P. malariae*: 22.4 [8.5–72.2] p/μL, n = 175; *P. ovale* spp.: 5.8 [2.0–28.0] p/μL, n = 71), vs. symptomatic clinic sub-population (*P. malariae*: 36.5 [4.7–182] p/μL, n = 132; *P. ovale*: 17.7 [4.6–65.8] p/μL, n = 93). Parasite density distributions were slightly higher among *P. malariae* infections than *P. ovale* spp. infections. We did not detect any differences in estimated *P. malariae* or *P. ovale* spp. parasite density distributions between mixed- vs. single-species *P. malariae* or *P. ovale* spp. infections in the total population (*P. malariae: p* = 0.071; *P. ovale: p* = 0.465),

**a.**

**b.**

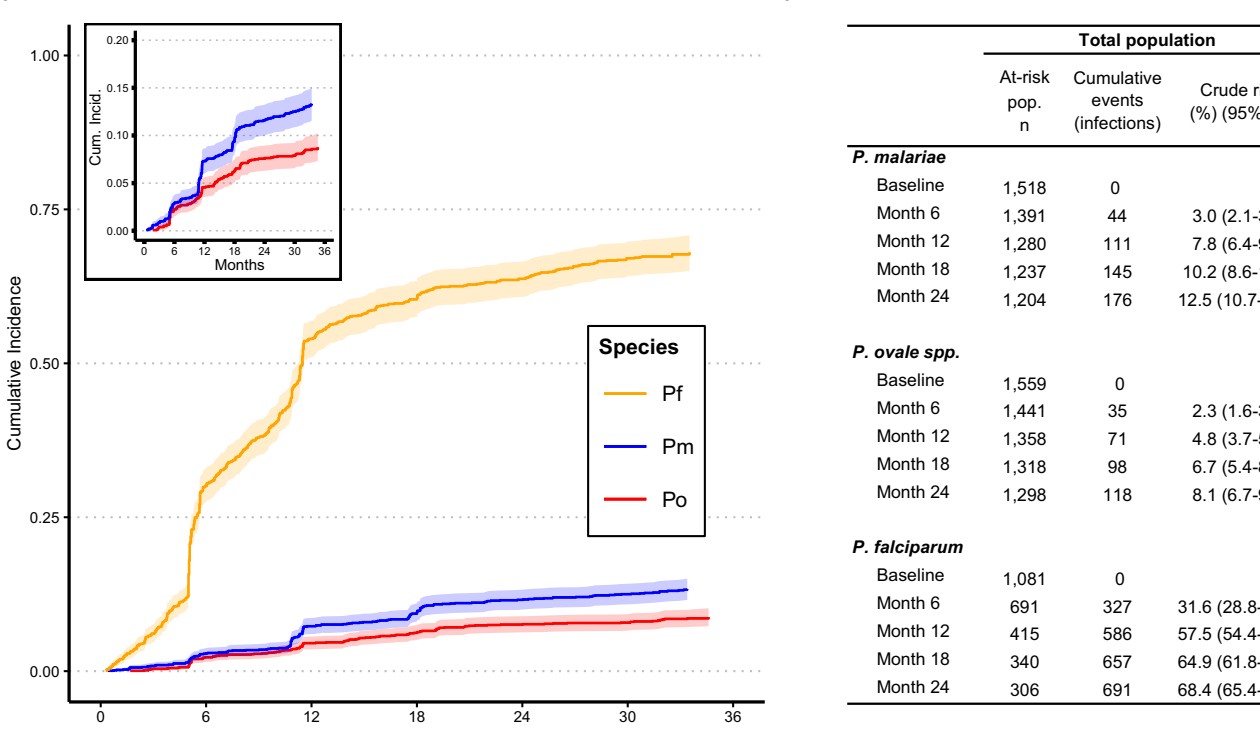

| | Total population | | |
|---|---|---|---|
| | At-risk pop. n | Cumulative events (infections) | Crude risk (%) (95% CI) |
| ***P. malariae*** | | | |
| Baseline | 1,518 | 0 | |
| Month 6 | 1,391 | 44 | 3.0 (2.1-3.9) |
| Month 12 | 1,280 | 111 | 7.8 (6.4-9.1) |
| Month 18 | 1,237 | 145 | 10.2 (8.6-11.8) |
| Month 24 | 1,204 | 176 | 12.5 (10.7-14.2) |
| ***P. ovale* spp.** | | | |
| Baseline | 1,559 | 0 | |
| Month 6 | 1,441 | 35 | 2.3 (1.6-3.1) |
| Month 12 | 1,358 | 71 | 4.8 (3.7-5.9) |
| Month 18 | 1,318 | 98 | 6.7 (5.4-8.0) |
| Month 24 | 1,298 | 118 | 8.1 (6.7-9.5) |
| ***P. falciparum*** | | | |
| Baseline | 1,081 | 0 | |
| Month 6 | 691 | 327 | 31.6 (28.8-34.4) |
| Month 12 | 415 | 586 | 57.5 (54.4-60.5) |
| Month 18 | 340 | 657 | 64.9 (61.8-67.7) |
| Month 24 | 306 | 691 | 68.4 (65.4-71.2) |

**Fig. 2 | Cumulative incidence of *Plasmodium* infections across 34-months of follow-up in the total study population. a** Cumulative incidence curves depict time to the first detected *P. malariae* (blue), *P. ovale* (red), and *P. falciparum* (yellow) infections among participants negative for each *Plasmodium* species infection at baseline. Shading depicts 95% CIs around cumulative incidence estimates. Incidence curves increase in a step-wise fashion at follow-up survey timepoints due to participant-wide screening and case detection, whereas gradual slopes in between survey timepoints indicate additional incident events detected through symptomatic presentation to clinics. **b** Crude risks of incident infections among the total population shown in Fig. 2a, by malaria species. Time to the first incident infection since baseline. CI confidence interval, Pf *Plasmodium falciparum,* Pm *Plasmodium malariae,* Po *Plasmodium ovale* spp.

or across specific population types. Parasite densities are summarized in Supplementary Table 5; samples rehydrated during molecular analysis were excluded from analysis (*n* = 591 total).

Interestingly, differences in the distribution of *P. falciparum* parasite densities were observed between single-species *P. falciparum* infections vs. mixed-species infections with *P. malariae*, but not mixed *P. ovale* spp. in the clinic subpopulation (Fig. 5). A higher median (IQR) *P. falciparum* parasite density was observed within mixed *P. falciparum- P. malariae* co-infections (127 [25–496] p/μL, *n* = 122; vs. 47 [7.4–311] p/μL, *n* = 1584; *p* = 0.001) and *P. falciparum- P. ovale* co-infections (122 [42.2–486] p/μL, *n* = 49; vs. 47 [7.4–311] p/μL, *n* = 1584; *p* = 0.01) as compared to single-species *P. falciparum* infections among the survey population. However, a lower median *P. falciparum* parasite density was observed within mixed *P. falciparum- P. malariae* infections (*n* = 84) compared to *P. falciparum* single-species infections (*n* = 1834) in the clinic-based sub-population (288 [57–4319] p/μL vs. 2897 [127–18,058] p/μL; *p* < 0.001); this association was not observed for *P. falciparum- P. ovale* co-infections (1710 [165–13,082] p/μL, *n* = 47; vs. 2897 [127–18,058] p/μL; *p* = 0.596), although the sample size for mixed *P. falciparum-P. ovale* co-infections was smaller than for single-species *P. falciparum* infections.

## Discussion

*P. malariae* and *P. ovale* spp. malaria infections were detected across all sites included in this study, affecting children and adults across a range of *P. falciparum* transmission intensities in Kinshasa Province, DRC. In the largest longitudinal study of non-falciparum malaria conducted to-date in the DRC, we observed an 8% cumulative incidence of first *P. malariae* infection and 5% incidence of first *P. ovale* spp. infection within 1-year, including both asymptomatic and symptomatic

infections. Over half of the study population in this high malaria-burdened country experienced an incident *P. falciparum* infection within 1-year from baseline, confirming the central importance of *P. falciparum* in the DRC. Over 70% of the non-falciparum infections detected occurred as co-infections with *P. falciparum*. Though *P. falciparum* was the dominant species, we detected a considerable risk of infection by a non-falciparum species, particularly for children under 15 years of age. These findings underscore the need to consider these neglected species in areas of high *P. falciparum* transmission.

Ten percent of household survey participants had at least one *P. malariae* infection during the study, and 5% had at least one *P. ovale* spp. infection, including both mixed- and single-species infections. Similar prevalences were observed among symptomatic individuals during clinic-based surveillance. Prevalences generally align with previous estimates of non-falciparum infections in the DRC derived from cross-sectional studies[13,15–21,30]. Infection prevalences were stable across the 34-month follow-up, differing from recent studies in neighboring countries, which have detected increases in *P. malariae* and *P. ovale* infection prevalence over time, particularly as *P. falciparum* prevalences have started to decline[3,4,6]. The high proportion of participants with *P. malariae* who had infection detected more than once during the study (24%) is notable given that this parasite can persist in the blood at low parasite density for long periods of time[25,31,32]. Chronic *P. malariae* infection may be associated with deleterious clinical outcomes, such as severe anemia and chronic nephrotic syndrome[24,25,33]. Though we could not distinguish whether multiple *P. malariae* infections detected in an individual were due to chronic occurrences or acute reinfection events, future work may allow us to differentiate these infection types. Such work is necessary given a dearth of evidence on the commonality of chronic *P. malariae* infection, as well as

**Table 2 | Incidence of infection during household survey visits and symptomatic clinic visits, by *Plasmodium* species**

| | Survey population | | | Clinic subpopulation | | |
|---|---|---|---|---|---|---|
| | At-risk pop. n | Cumulative events (infections)[a] | Crude risk (%) (95% CI) | At-risk pop.[b] n | Cumulative events (infections)[a,c] | Crude risk (%) (95% CI) |
| *P. malariae* | | | | | | |
| Baseline | 1518 | 0 | | 1002 | 0 | |
| Month 6 | 1398 | 20 | 1.4 (0.8–2.1) | 934 | 24 | 2.5 (1.5–3.5) |
| Month 12 | 1230 | 67 | 4.9 (3.7–6.0) | 910 | 48 | 5.0 (3.6–6.4) |
| Month 18 | 744 | 86 | 7.0 (5.5–8.4) | 890 | 68 | 7.1 (5.5–8.7) |
| Month 24 | -- | -- | -- | 875 | 83 | 8.7 (6.9–10.4) |
| *P. ovale* spp. | | | | | | |
| Baseline | 1559 | 0 | | 1001 | 0 | |
| Month 6 | 1425 | 23 | 1.6 (1.0–2.3) | 990 | 12 | 1.2 (0.5–1.9) |
| Month 12 | 1279 | 49 | 3.4 (2.4–4.3) | 972 | 30 | 3.0 (1.9–4.0) |
| Month 18 | 769 | 55 | 4.1 (3.0–5.1) | 951 | 51 | 5.1 (3.7–6.4) |
| Month 24 | -- | -- | -- | 938 | 64 | 6.4 (4.9–7.9) |
| *P. falciparum* | | | | | | |
| Baseline | 1081 | 0 | | 662 | 0 | |
| Month 6 | 774 | 227 | 22.5 (19.9–25.0) | 520 | 143 | 21.6 (18.4–24.6) |
| Month 12 | 529 | 423 | 42.8 (39.6–45.8) | 323 | 341 | 51.4 (47.5–55.1) |
| Month 18 | 167 | 458 | 51.4 (47.6–55.0) | 238 | 425 | 64.1 (60.3–67.6) |
| Month 24 | -- | -- | -- | 211 | 452 | 68.2 (64.4–71.5) |

*CI* confidence interval.
[a]Cumulative first infections detected. Time to the first incident infection event.
[b]The at-risk population for the clinic subpopulation comprises subjects who were PCR-negative for each species-specific infection at baseline and had at least one visit to study health clinics during the follow-up period.
[c]Includes infections detected only at symptomatic clinic visits.

reinfection rates for non-falciparum species in regions highly endemic for *P. falciparum*.

Populations experiencing a *P. malariae* or *P. ovale* spp. infection were similar but not identical to those infected with *P. falciparum*. School-aged children were found to have higher prevalences of *P. malariae* infection than children under 5 years of age, following similar age-related infection associations with *P. falciparum* that have been previously established in the region[34,35]. Dissimilar from *P. falciparum* infection patterns, symptomatic children under 5 had comparable *P. malariae* infection prevalences as those experienced by symptomatic adults aged 15 and older. All *Plasmodium* species were generally more common in rural and peri-urban sites, and less common in the wealthiest households.

We observed a crude association between *P. malariae* infection prevalence and anemia of any severity in the symptomatic clinic sub-population, which translated to a 1.5% absolute increase in prevalence compared to those without anemia. Although, this association was no longer significant when stratifying by anemia severity, and was not detected among *P. ovale* spp. infections in this study. Similarly, no association between non-falciparum infections and fever was detected within the survey- or clinic-based analyses, though misclassification of fever is possible if fever-reducing medicines were taken prior to clinic visits, as data on use of fever-reducing medicines was not captured in this study. Prior studies have posited a protective effect of co-infection with *P. falciparum* and non-falciparum species on reduced clinical outcomes, compared to single-species *P. falciparum* infections[36,37], although this has not been consistently observed across studies[38,39]. We did not assess causal effects of non-falciparum infection on clinical outcomes in our study due to the limited number of positive samples (though no crude associations were detected in the unadjusted models), including the possibility of attenuated *P. falciparum* symptoms due to mixed-species infection.

Interestingly, *P. falciparum* parasite densities were higher in mixed infections with *P. malariae* or *P. ovale* spp. in the household survey-based, but not clinic-based, analysis. One possible explanation is that *P. falciparum* increases its virulence to compete with non-falciparum species within the asymptomatic host, though this is an unproven hypothesis that our data are insufficient to evaluate. Non-falciparum parasite densities observed in this study were low, as is characteristic of *P. malariae* and *P. ovale* spp. infections. We did not identify any difference in non-falciparum parasite density between mixed and single-species non-falciparum infections.

Our incidence and prevalence estimates should be considered lower bounds of the true incidence and prevalence in the study catchment area owing to several limitations of the study. First, the biannual frequency of household surveys missed infections that occurred and resolved between study visits. Also, while the long-itudinal design provides helpful insight into risk of infection surrounding these species, the study's sample size and duration of the study likely limited our ability to detect changes in the prevalence of these less-common species over time in the DRC, where gains in malaria control have not been achieved as in other settings. Second, while loss to follow-up was low overall in the study, missed study visits, early loss to follow-up, and differences in study clinic attendance across sites may have introduced selection bias into the study. Parti-cipants may have elected to visit alternative health facilities for malaria care instead of presenting to study clinics. However, all participants were eligible for free malaria treatment at study clinics to encourage use of the study facilities for care and increase representativeness of the clinic subpopulation to the overall study population. Third, our duplex PCR assay detected low density infections with a limit of detection between 1 and 10 p/μL, but would likely miss infections of parasite densities below this range, which are known to occur with *P. malariae* and *P. ovale* species. Fourth, some misclassification of sea-sonality may be present, as local classifications for seasonality by month, rather than rainfall amounts, were used to define rainy seasons. Finally, our observational study design is not immune to the risk of unmeasured confounding, although no causal relationships were

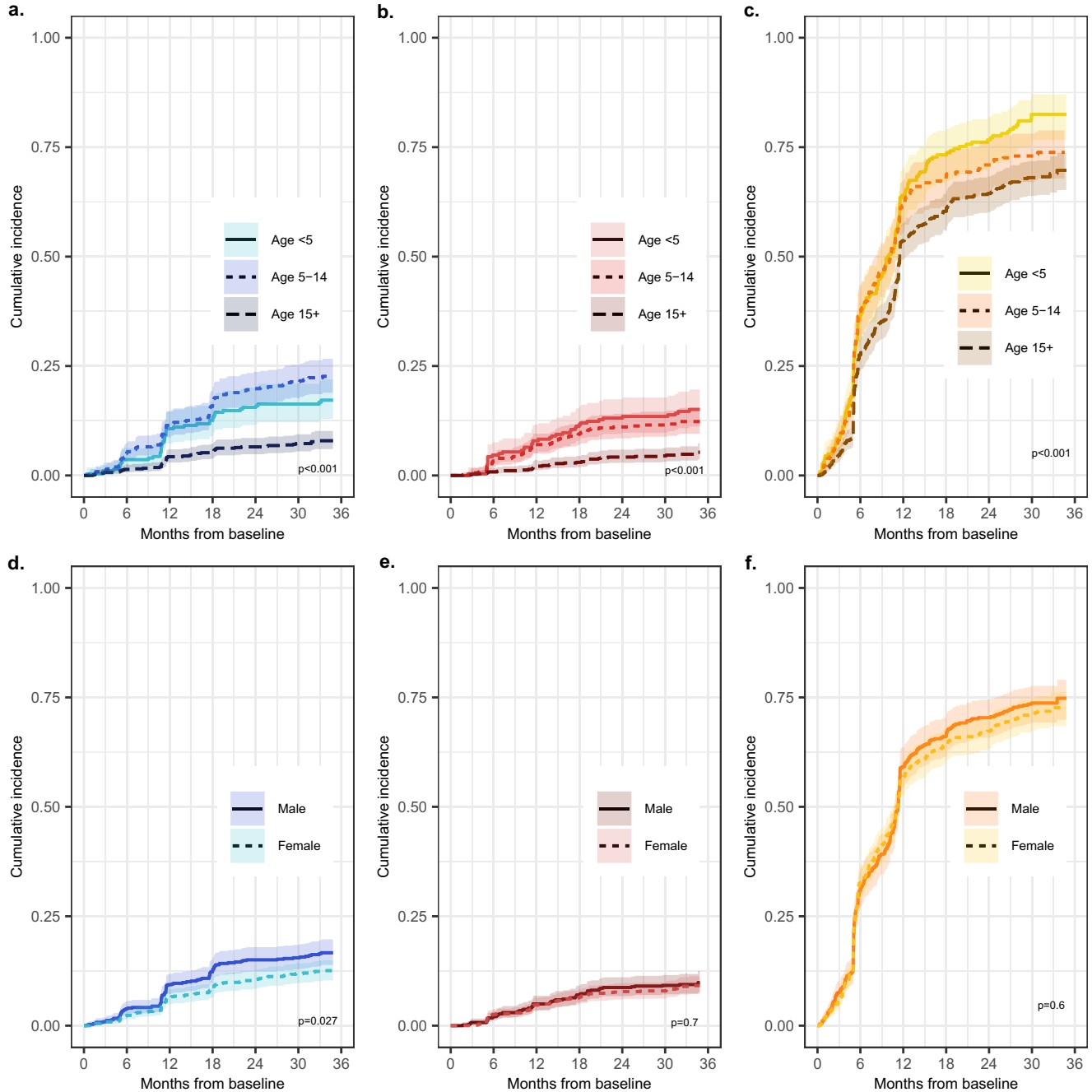

**Fig. 3 | Cumulative incidence curves stratified by participant characteristics.** Cumulative incidence of first infection stratified by age at baseline, for *P. malariae* (*n* = 1518; <5 years: *n* = 296, 5–14 years: *n* = 470; 15+ years: *n* = 752 at risk at baseline) in blue (**a**), *P. ovale* spp. (*n* = 1557; <5 years: *n* = 301, 5–14 years: *n* = 498; 15+ years: *n* = 758 at risk at baseline) in red (**b**), and *P. falciparum* (*n* = 1081; <5 years: *n* = 238, 5–14 years: *n* = 260; 15+ years: *n* = 583 at risk at baseline) in yellow (**c**). Cumulative incidence of first infection stratified by sex, for *P. malariae* (*n* = 1518; female: *n* = 840, male: *n* = 678 at risk at baseline) in blue (**d**), *P. ovale* spp. (*n* = 1559; female: *n* = 858, male: *n* = 701 at risk at baseline) in red (**e**), and *P. falciparum* (*n* = 1081; female: *n* = 618 male: *n* = 463 at risk at baseline) in yellow (**f**). *p*-values are calculated from log-rank testing. Shading depicts 95% CIs around incidence estimates. CI confidence interval.

assessed in this study. Associated factors depict crude associations only, and while the sample size was relatively large, findings may be affected by the lower prevalence of *P. malariae* and *P. ovale* events.

This study provides insight into the epidemiology of *P. malariae* and *P. ovale* species in a region heavily affected by *P. falciparum* malaria. Though less common and impactful than *P. falciparum*, *P. malariae* and *P. ovale* spp. infections occurred across all sites and subpopulations, and were often detected within symptomatic clinic-based cases in this large longitudinal study in the DRC. Malaria research and control efforts focused on *P. falciparum* should also

consider these neglected species, particularly in school-aged children and rural communities as part of efforts to progress toward malaria elimination.

## Methods

### Study design
This study leverages data and samples collected during a longitudinal cohort study of malaria transmission across seven sites within three health areas of varying urbanicity and malaria endemicity in Kinshasa Province, DRC between 2015 and 2017. Detailed methods for site and

a.

b.

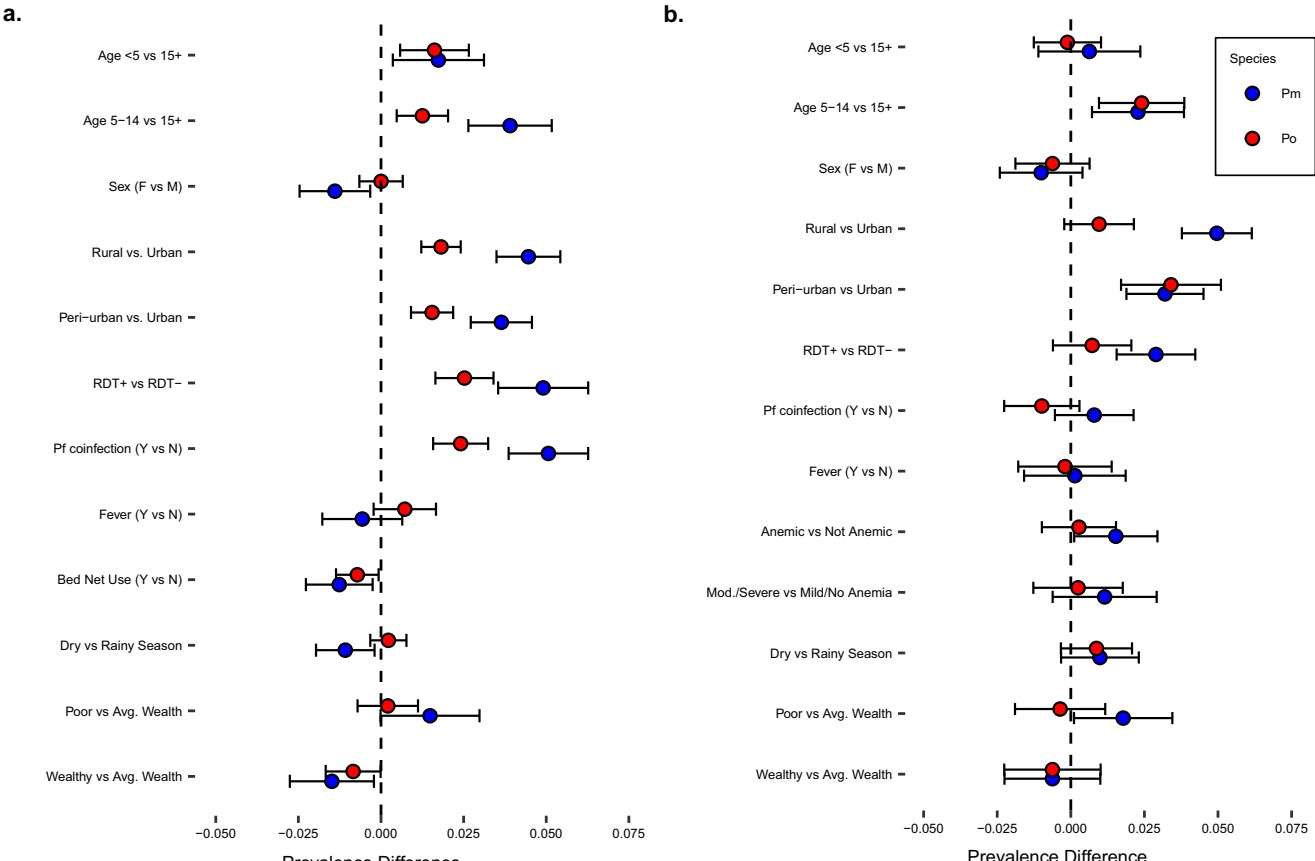

**Fig. 4 | Factors associated with *P. malariae* and *P. ovale* spp. infection prevalence, stratified by survey vs. clinic-based population. a** Factors associated with *P. malariae* (blue) and *P. ovale* (red) infection prevalences at survey visits (baseline and three follow-up surveys). **b** Factors associated with *P. malariae* (blue) and *P. ovale* (red) prevalences at symptomatic clinic visits across follow-up (i.e., symptomatic cases). Error bars represent 95% CIs around unadjusted prevalence difference measures; numerical values of prevalence differences and 95% CIs are listed in Supplementary Tables 8 and 9. *n* = 5659 biologically independent samples in the survey population (**a**), and *n* = 3349 in the clinic-based population (**b**). Factors associated with non-falciparum infection prevalences compared to *P. falciparum* infection are depicted in Supplemental Fig. 2. CI confidence interval, Pf *Plasmodium falciparum,* Pm *Plasmodium malariae,* Po *Plasmodium ovale* spp., RDT rapid diagnostic test.

household sampling have been previously described[34,40]. In brief, households were selected at random within the seven chosen villages, and household members were enrolled into the study following screening for eligibility criteria and provision of informed consent, or assent for minors. Participants were followed prospectively through biannual household visits ('survey-based population'; active surveillance), and at visits to local study health facilities, as-needed, for presentation of fever or other malaria symptoms ("clinic-based subpopulation"; symptomatic passive surveillance) (Fig. 6). Three local health facilities were included in our study, one per health area. The chosen facility was the main government clinic in two of these health areas (Bu – rural, Kimpoko – peri-urban). In urban Voix de Peuple, one private clinic was selected due to the diverse patient population it served. Only households within the clinic catchment areas that were located within 5 km from the clinic were eligible for the study.

Household surveys were conducted between February 2015 and October 2016 (20 months of active surveillance), with baseline and second follow-up surveys largely occurring during the rainy season, and first and third (final) follow-up surveys occurring during the dry season. Symptomatic clinic visits continued beyond the completion of household surveys, through December 2017, for 34 months of passive surveillance.

At baseline and follow-up surveys, household-level and individual-level questionnaires were collected to ascertain demographic information, household characteristics (e.g., housing materials, possessions for wealth indicators, bed net ownership and use), health status and clinical data (e.g., malaria diagnostic history, recent symptoms, treatment use). All participants were screened for malaria infection at each visit using a combination RDT detecting *P. falciparum*-specific HRP2 and pan-*Plasmodium* LDH) antigens (SD Bioline Ag P.f./Pan RDT [05FK60], Alere, Gyeonggi-do, Republic of Korea). Those positive by RDT (defined as a positive result for either antigen) were referred to study clinics for antimalarial treatment. DBS samples were also collected from all consenting participants at each visit using Whatman 3MM filter paper (Fisher Scientific, Fair Lawn, NJ USA), and were stored with desiccant at -20 °C for future molecular testing.

Brief clinical questionnaires were collected from participants at each symptomatic clinic visit throughout the study, alongside malaria RDTs, DBS samples, and hemoglobin testing. Participants were treated at health facilities according to local clinical judgment, as needed, including artemisinin-based combination therapy (ACT) for malaria infection, or referral for blood transfusion if severely anemic.

**Plasmodium species determination**

Samples underwent molecular testing at the University of North Carolina Chapel Hill to distinguish *Plasmodium* species. DNA was extracted from DBS samples using Chelex and saponin[41]. *P. falciparum* parasitemia was identified using a quantitative real-time duplex PCR assay targeting the LDH and human β-tubulin genes, as previously described[40–42]. We tested samples for *P. malariae* and *P. ovale* spp. parasitemia using a semi-quantitative, duplex real-time PCR assay targeting the 18S ribosomal subunit (Supplementary Table 6)[43,44]. A

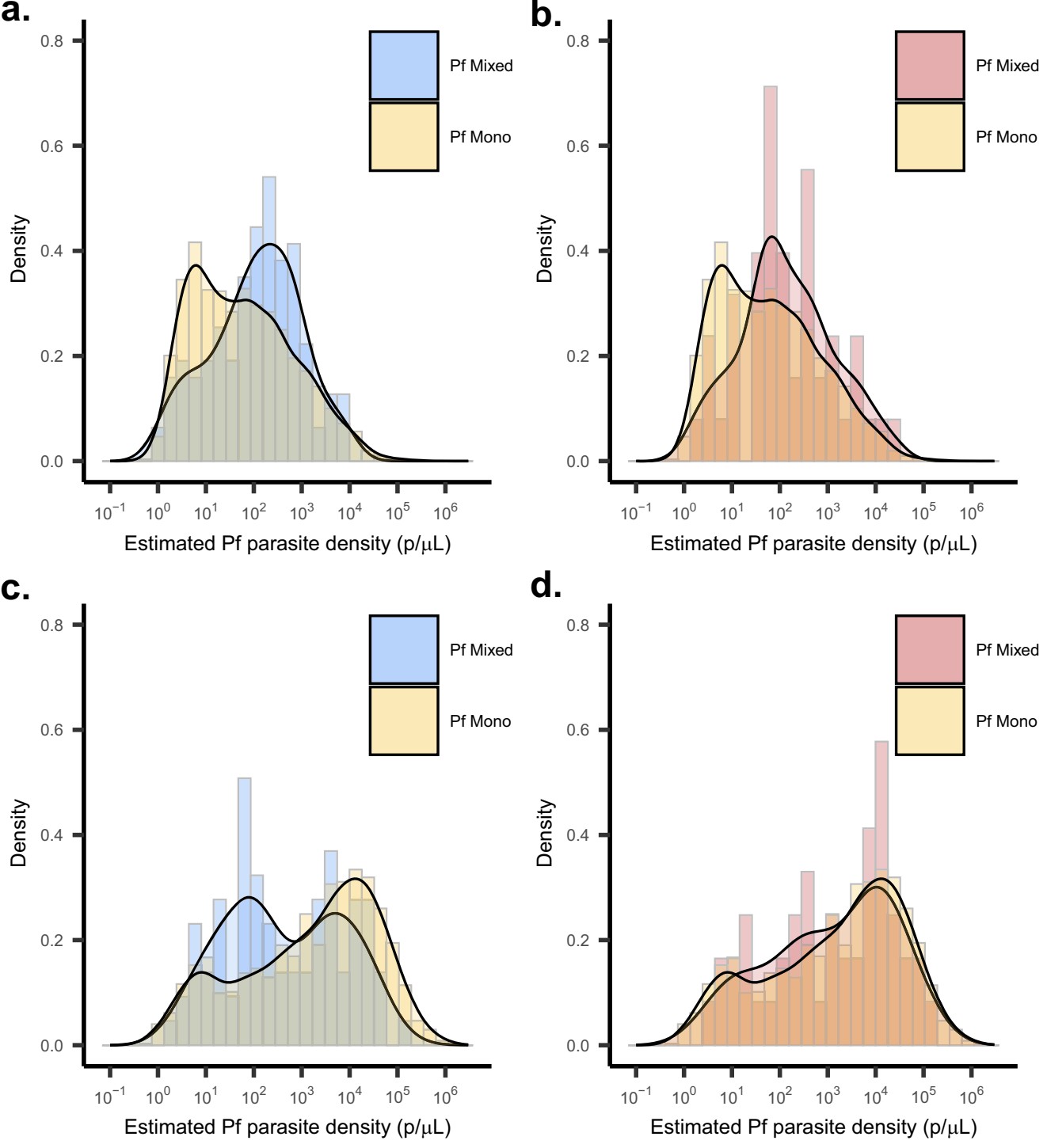

**Fig. 5 | Distributions of *P. falciparum* parasitemia by type of non-falciparum mixed infection vs. *P. falciparum* mono infections, stratified by population.** Household survey-based results for *P. falciparum* parasitemia distributions comparing *P. falciparum* mono infections (*n* = 3418) in yellow vs. co-infection with (**a**) *P. malariae* (*n* = 206) in blue and (**b**) *P. ovale* spp. (*n* = 96) in red. Clinic sub-population

*P. falciparum* parasitemia distributions comparing *P. falciparum* mono infections (*n* = 1834) in yellow vs. co-infection with (**c**) *P. malariae* (*n* = 84) in blue and (**d**) *P. ovale* spp. (*n* = 47) in red. Density distributions are displayed to illustrate trends in overall distributions accounting for variation in sample sizes across groups. Pf *Plasmodium falciparum*.

single replicate of non-falciparum PCR assays were run on samples and considered positive if amplification occurred at cycle thresholds (Cts) < 40. *P. falciparum* samples were run in duplicate and were considered positive if at least one sample amplified at a Ct <38, or both amplified between Cts ≥38 and <40, as previously described[45]. Semi-quantitative *P. malariae* and *P. ovale* spp. parasite densities were estimated from standard curves made using serially-diluted 18 S rRNA

plasmid DNA (BEI Resources: #MRA-179 and #MRA-180), assuming six plasmid copies per parasite genome[46]. Non-falciparum parasitemias were then multiplied by 4.0 to account for a 4-fold dilution of blood from three DBS punches[47], improving comparability with *P. falciparum* parasitemia quantification. DNA samples with evidence of dehydration were rehydrated prior to PCR testing using 50 μL of molecular-grade water; rehydrated samples were excluded from parasite density

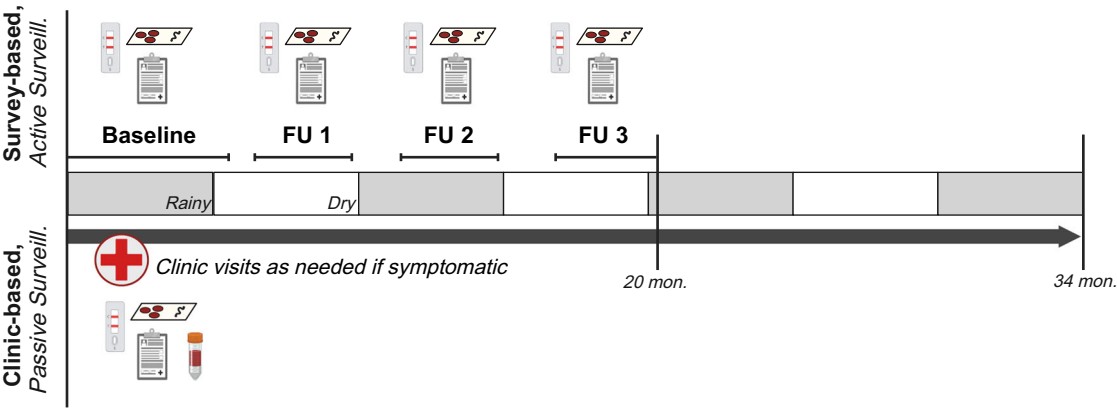

**Fig. 6 | Study design overview**[51]. Biannual household study visits occurred at baseline and three follow-ups over a 20-month longitudinal period in which household demographics and individual health surveys were collected, malaria RDTs were performed, and DBS were collected. Participants who developed fever or other malaria symptoms during follow-up visited study health clinics, where clinical questionnaires, malaria RDTs, a DBS, and hemoglobin testing were performed. Study participants completing household surveys were included in the survey population; those who also visited the clinic at least once for malaria symptoms were included in the clinic subpopulation. DBS dried blood spot, RDT rapid diagnostic test.

analyses. Negative (water) controls were included on all PCR runs. *P. vivax* was not tested in this study as previously published estimates of *P. vivax* prevalence in Kinshasa Province from a similar time period (2013-14) indicated low prevalence in the region (0 to <2%)[30].

## Population

All participants with a baseline DBS available for testing were included in the study. The Kinshasa longitudinal cohort study population provided two distinct types of data utilized in this study: (1) The "survey-based" analysis consists of participant survey data and clinical samples collected at active surveillance household visits for all those enrolled at baseline, and (2) the "clinic-based" analysis consists of participant clinical survey data and samples collected from the subset of the survey-based population who had samples collected at the baseline household enrollment visit, and who visited study clinics as-needed when symptomatic during follow-up (passive surveillance).

Samples derived from all study touchpoints (household surveys + clinic visits) were combined to estimate overall infection burden, and also stratified by population type (survey vs. clinic) to account for differences between their predominantly asymptomatic vs. symptomatic natures. Characteristics between the broader survey population and the nested clinic subpopulation were assessed (Supplementary Table 1). Weighting was not performed as factors driving selection into the clinic subpopulation were expected to be directly related to the probability of the outcome.

## Data analysis

The primary outcomes in this study were *P. malariae, P. ovale* spp., and *P. falciparum* infection confirmed by real-time PCR. Demographic and clinical characteristics were summarized by *Plasmodium* species across study populations, and the incidence and prevalence of infections were estimated across the 34-month study follow-up period. Parasite densities were descriptively summarized by species and infection type (mixed vs. single-species).

Cumulative incidences of first *P. malariae, P. ovale* spp., and *P. falciparum* infections were calculated using Kaplan-Meier methods among participants who were PCR-negative at baseline for the *Plasmodium* species of interest. Incidences were estimated as the inverse of the survival function, measuring time (in months) to the first detected infection since baseline. Participants in the survey population were censored at the time of their last study household visit. Participants in the clinic subpopulation were censored at the end of passive surveillance follow-up, as participants were eligible to continue visiting study health clinics even after the final active follow-up, if

symptomatic. Similarly, in the total population, participants who were lost to follow-up prior to the final household survey and had never visited study clinics during follow-up were censored at the time of their last study household visit. Otherwise, those in the total population who completed the final household follow-up were censored at the end of the full study period. Participants dropped out of the at-risk population at the time of their first infection event. We assumed no competing risks for malaria infection since infection by one species would not preclude infection by another, and measured mortality events in the study were low ($n = 26$). Demographic characteristics associated with incidence of first infection were evaluated by log-rank testing of stratified cumulative incidence curves.

Prevalences of *P. malariae, P. ovale* spp., and *P. falciparum* were estimated overall and by species across follow-up. Factors associated with non-falciparum infection prevalences were evaluated using crude linear binomial generalized estimating equation (GEE) models accounting for repeated testing of participants over time. Possible factors associated with malaria infection were modeled individually. An exchangeable working correlation matrix was assumed for GEE models, and robust standard errors were estimated for calculation of 95% confidence intervals (CIs). While three-level clustering of participants nested within households, and households within village sites is present in the study, this clustering structure was too complex to account for using mixed-effect modeling because of the low number of *P. malariae* and *P. ovale* outcomes detected in this study. Therefore, the GEE model was used. We also assessed the frequency of multiple same-species non-falciparum infections detected within a participant throughout follow-up; however, these were not distinguished between acute reinfection events, or chronic carriage of a prior infection.

Household wealth was computed through adaptation of the Demographic and Health Survey method and categorized into quintiles as previously described[48]. Quintiles were collapsed into three categories for analysis, and wealth at baseline was carried forward for all subsequent study visits. Seasonality was defined by month, classifying October through April as the rainy season. Anemia severity at symptomatic clinic visits was classified according to hemoglobin (Hb) level, following WHO categories for age-, sex-, and pregnancy-specific cut-points[49]; severe anemia was defined as Hb<7.0 g/dL for children <5 years, and Hb <8.0 g/dL for all others. Cases were categorized as any anemia vs. no anemia, and comparing moderate/severe vs. mild/no anemia cases. Fever was defined as an axillary temperature >37.5 °C as measured at symptomatic clinic visits, or as self-reported fever within the prior week at baseline and follow-up surveys. Missing data were summarized and excluded from statistical analyses.

Dataset construction and cleaning were performed using SAS (version 9.4), and analyses were conducted in R (version 4.0.2) primarily using *tidycmprsk, ggsurvfit*, and *gee* packages. The full list of R packages used in analysis is listed in Supplemental Table 7. Chi-square and Fisher's Exact tests were performed for statistical comparison of categorical variables. Kruskal-Wallis rank sum tests were performed for comparison of continuous variables, including parasite density distributions across species, assuming nonnormality. All *p*-values were two-sided. Informed consent, and assent where required, was obtained from all participants or their legal guardians prior to study enrollment and sampling. The study was approved by the Institutional Review Boards at the University of North Carolina at Chapel Hill (IRB#: 14-0489) the University of Iowa (#201701201), and the Kinshasa School of Public Health (ESP/CE/015/014).

## Reporting summary
Further information on research design is available in the Nature Portfolio Reporting Summary linked to this article.

## Data availability
The de-identified analysis dataset is publicly available through the Carolina Digital Repository at: https://doi.org/10.17615/kjjg-7a88.

## Code availability
Analysis R code is publicly available at: https://github.com/IDEELResearch/nonfalciparum_in_DRC[50].

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

## Acknowledgements

We thank the study team for conducting household surveys and field sampling, as well as all study participants for their time and engagement throughout multiple years of follow-up. We also thank the late Prof. Steven Meshnick for his leadership in establishing the parent study and for mentorship to DRC- and US-based co-authors. Lastly, we thank Seungwon Kim at the University of Iowa for their calculation of wealth scores by principal component analysis (PCA) provided for this analysis. This study was funded in part by the National Institutes of Health (NIH) [R21 AI148579 to J.B.P. and J.T.L.; R01 AI139520 to J.A.B.; R01AI132547 and K24AI134990 to J.J.J.; R01AI107949 and R01AI129812 to A.T.; T32AI070114 to R.S. and K.B.; D43TW009340 to K.B.], and by an ASTMH/Burroughs-Wellcome Fund award to J.B.P. The funders had no role in study design, data collection and analysis, decision to publish, or preparation of the manuscript. The following reagents were obtained through BEI Resources, NIAID, NIH: Diagnostic Plasmid Containing the Small Subunit Ribosomal RNA Gene (18S) from *Plasmodium malariae*, MRA-179, and *Plasmodium ovale*, MRA-180, contributed by Peter A. Zimmerman. The following reagent was obtained through BEI Resources, NIAID, NIH: *Plasmodium falciparum*, Strain 3D7, MRA-102, contributed by Daniel J. Carucci.

## Author contributions

R.S., K.B. and J.B.P. conceptualized and designed the study. R.S., M.S. and K.L.T. performed laboratory analyses. R.S., K.B., M.M.K., N.M., M.N., G.K., J.A., Jo.A.B., K.L.T., W.M.S., V.G., T.N. and J.B.P. performed data collection and data cleaning. R.S., K.B., V.G., M.E., M.C., Je.A.B., J.J.J., A.T., J.T.L. and J.B.P. conducted data analyses and interpreted results. R.S., K.B., J.B.P. wrote the manuscript draft. All authors reviewed and approved the manuscript.

## Competing interests

J.B.P. reports research support from Gilead Sciences, non-financial support from Abbott Laboratories, and consulting for Zymeron Corporation, all outside the scope of this study. The remaining authors declare no competing interests.
