## [Peer Review File · Nature Communications]

Epidemiology of Plasmodium malariae and Plasmodium ovale spp. in Kinshasa Province, Democratic Republic of CongoREVIEWER COMMENTS

Reviewer #1 (Remarks to the Author):

This is a thoroughly conducted and well-reported study of non-falciparum malaria in an important high-burden sub-Saharan African setting. The methods and results are shared clearly and the extension of the analysis beyond reports of incidence/prevalence into the varying presence of mixed and mono infections in different populations and the differences in parasite densities make this a useful contribution to the field.

Major comments

1. Reporting non-falciparum epidemiology without any mention of vivax malaria (beyond citations) was surprising. It would be useful to know why vivax malaria was not included in this study.

Minor

1. Line 99 – *P. ovale* acute symptoms given ability to relapse?
2. Line 134 – was net use as well as ownership assessed?
3. Starting line 138 – clarify if DBS was taken for all or just those RDT pos?
4. Around line 188 – clarify if/that the clinical catchment is the same as the sampled communities and how that is known
5. Line 401 – citation for malariae known recrudescence? More so that *ovale* relapse?
6. Line 443 – and clinics were easily physically accessible to community members?
7. Line 445 – what is very low density?

Reviewer #2 (Remarks to the Author):

Nature Communications Epi of Pm and Po in DRC review

This is a well written article summarizing a basic yet useful epidemiological study of the prevalence and incidence of non-falciparum and falciparum infections over three years in the DRC. A few critical methodological details should be clarified, including the justification for calculation of period prevalence and the overall sampling.

Methods

It would be helpful to provide a bit more detail on the sampling, even though the authors note these are described in further detail elsewhere. For example, how were the villages selected? Was probability sampling used allowing for population-based estimates? Additionally, were visits at health facilities only from enrolled participants at the baseline household visit included? It appears this is the case but it would be very helpful to clarify these details more explicitly.

Page 9: "Prevalences of *P. malariae*, *P. ovale* spp., and *P. falciparum* were estimated overall, and by infection type, calculated as the number of infections detected out of the total number of visits in the study". Do you mean to say that "Prevalence of Plasmodium infection was estimated overall, and by species type (*P. malariae*, *P. ovale* spp., and *P. falciparum*)...."?

Also, this is a rather unusual definition of period prevalence. Period prevalence is typically defined as the number of people with the disease/condition at any time during the period. To me it does not make sense to use the number of visits as the denominator—why not simply the number of people included during the time period and the numerator is any infection during that time period? Please justify the unusual methodology used to calculate period prevalence.

Also please clarify how the differences in prevalences were calculated. Page 9 of the methods indicates that binomial GEE was used but was one adjusted model used? Or was each risk factor (presented in Figure 5) evaluated on its own? Please add this detail.

Discussion

Pages 17-18: "The high proportion of *P. malariae*-infected participants who experienced multiple infections in the study (24%) is notable given that this species is known to recrudescence." What evidence is there for recrudescence of *P. malariae* infections? Vivax and *ovale* infections are known to relapse as they have dormant liver stages, but *malariae* infections do not have hypnozoites. Please elaborate on this statement and provide references for the assertions made.

Table 2: The title of the table is slightly misleading; it only mentions non-falciparum infection but falciparum infection is also presented. Can you amend to include mention of falciparum infection in the title please?

Table 2: How is the baseline (and year 1 and year 2) at-risk population defined for the clinic sub-population? Are these participants who were negative for the malaria species at the baseline household visit AND who had at least one clinic visit during the follow-up period? Or is this otherwise defined? Please add more details to footnote 2 for the clinic sub-population.

Table 2 and Figure 3: I am having a hard time understanding how the number of new infections and the at-risk population compares between Table 2 and Figure 3. The numbers of infections do not seem to quite add up to the household plus clinic population. There are no household events reported during year 2 and the number of new infections during year 2 in panel b of Figure 2 does not equal those reported in Table 2. Please clarify.

Supplemental Table 3: Are the characteristics included in this table those of the participants at the baseline household visit? And the PCR positive are those with any positive PCR at the clinic at any time during the follow up period? Some clarifying notes would be helpful.

Nature Communications - Response to Reviewers

Manuscript #: NCOMMS-23-14691A

Title: Epidemiology of *Plasmodium malariae* and *Plasmodium ovale* spp. in Kinshasa Province, Democratic Republic of Congo

Responses to Reviewer Comments:

*Note – page numbers for revisions refer to the revised, tracked-changes manuscript draft. **Bold, underlined text** indicates new additions.*

In addition to changes made in response to reviewer feedback, we have incorporated several additional wording edits throughout the revised manuscript for improved clarity, alignment with required formatting specifications, grammatical changes, and updates as needed. These are not explicitly noted below for simplicity.

Reviewer #1 (Remarks to the Author):

This is a thoroughly conducted and well-reported study of non-falciparum malaria in an important high-burden sub-Saharan African setting. The methods and results are shared clearly and the extension of the analysis beyond reports of incidence/prevalence into the varying presence of mixed and mono infections in different populations and the differences in parasite densities make this a useful contribution to the field.

Major comments

1. Reporting non-falciparum epidemiology without any mention of vivax malaria (beyond citations) was surprising. It would be useful to know why vivax malaria was not included in this study.

Author Reply:

Thank you for this comment. We did not test for *Plasmodium vivax* in this study since another study previously published by our group found a low prevalence of *P. vivax* in the same region of Kinshasa, DRC, using Demographic and Health Survey data from a similar time period, 2013-14 (Brazeau N et al., *Nature Communications*, 2021. doi:10.1038/s41467-021-24216-3). Risk factors of *P. vivax* infection were also assessed in this prior study.

We agree that we should better acknowledge these previously reported *P. vivax* prevalence estimates in our manuscript for context and comprehensiveness. We have updated our manuscript accordingly.

Revisions: (Pg. 8). **“*P. vivax* was not tested in this study as previously published estimates of *P. vivax* prevalence in Kinshasa Province from a similar time period (2013-14) indicated low prevalence in the region (0 to <2%)³⁹.”**

Minor comments:

1. Line 99 – *P. ovale* acute symptoms given ability to relapse?

Author Reply: We have edited the manuscript to also include the clinical impacts attributed to *P. ovale* spp. infections.

Revisions: (Pg. 5), **“*P. ovale* spp. infections commonly result in low morbidity, although severe complications have been documented in case reports^{28,29}.”** The clinical relevance of *P. falciparum* co-infection with non-falciparum species is also unclear...”

2. Line 134 – was net use as well as ownership assessed?

Author Reply:

Bed net use (defined by participant self-report that they slept under a bed net the prior night) was assessed in risk factor models and reported in study results. We also measured and reported household bed net ownership, finding that 72.5% of participants at the baseline survey lived in a household that owned at least one bed net, while only 45.0% of the population reportedly slept under a bed net the prior night. Both metrics, bed net use and ownership, are reported for the survey population in Table 1.

We added minor edits to the manuscript to better highlight net ownership and use in the text.

Revisions: (Pg. 7), “At baseline and follow-up surveys, household-level and individual-level questionnaires were collected to ascertain demographic information, household characteristics (e.g., housing materials, possessions for wealth indicators, bed net ownership **and use**), health status and clinical data...”

3. Starting line 138 – clarify if DBS was taken for all or just those RDT pos?

Author Reply:

Dried blood spots were collected for all participants, regardless of RDT result, at each baseline and follow-up survey, and also at each unscheduled symptomatic health clinic visit. We edited the manuscript to clarify this point.

Revisions:

(Pg. 7), “**All** participants were screened for malaria infection at each visit using a combination RDT....”

(Pg. 7), “~~Where consented,~~ Dried blood spot (DBS) samples were also collected **from all consenting participants** at each visit using Whatman 3MM filter paper (Fisher Scientific, Fair Lawn, NJ USA), and were stored with desiccant at -20°C for future molecular testing...”

4. Around line 188 – clarify if/that the clinical catchment is the same as the sampled communities and how that is known

Author Reply: Thank you for this comment. All households were located within the clinical catchment areas of the study clinics. For two of the included health areas, Bu (rural), and Kimpoko (peri-urban) the study clinics were the main clinics in each health area. Each of the villages in those two health areas were selected for the study because of their proximity to the study clinics to ensure they were included in the clinic catchment or within 5km distance away. For the urban health area, Voix de Peuple, more health clinics were available to participants, so one neighborhood was selected in the catchment of a study clinic. This clinic served a socioeconomically diverse segment of the neighborhood’s population.

A total of 3 health clinics were included in the study.

We edited the manuscript to clarify the sampling and study clinics included.

Revisions: (Pg. 6), “**Three local health facilities were included in our study, one per health area. The chosen facility was the main government clinic in two of these health areas (Bu – rural, Kimpoko – peri-urban). In urban Voix de Peuple, one private clinic was selected due to the diverse patient population it served. Only households within the clinic catchment areas that were located within 5km from the clinic were eligible for the study.**”

5. Line 401 – citation for malariae known recrudescence? More so that ovale relapse?

Author Reply: Thank you for this comment. This was an error on our part. We have edited our manuscript to characterize the ability of *Plasmodium malariae* to persist long-term at low parasite densities in the blood. Citations have also been added.

Revisions:

(Pg. 20), “The high proportion of **participants with *P. malariae*** infected participants who experienced ~~had~~ multiple infections **infection detected more than once during the study** (24%) is notable given that this **parasite** species can **persist in the blood at low parasite density for long periods of time**^{25,42,43}. Chronic *P. malariae* infection may be associated with deleterious clinical outcomes...”

(Pg 20), “Though we could not distinguish whether multiple *P. malariae* infections **detected in an individual** were **due to** chronic occurrences or acute re-infection events...”

6. Line 443 – and clinics were easily physically accessible to community members?

Author Reply: All study health clinics were near main roads and walkable from enrolled households. Villages included as sites in the study were selected so that households would be no more than 5km from the clinic. However, differences in household wealth, ownership or access to transportation, work status and care-taking responsibilities could impact differences in accessibility to the clinic for reasons other than distance.

We have edited the manuscript slightly to note this.

Revisions: (Pg. 6), “**Only households within the clinic catchment areas that were located within 5km from the clinic were eligible for the study.**”

7. Line 445 – what is very low density?

Author Reply: Thank you for this comment. We agree this is a subjective term and we have edited the manuscript to more concretely characterize low and very low parasite densities.

Revisions: (Pg. 22), “Third, our duplex PCR assay detected low density infections, **with a limit of detection between 1-10 p/μL**, but would likely miss **infections of parasite densities below this range, which are known to occur with *P. malariae* and *P. ovale* species.**”

Reviewer #2 (Remarks to the Author):

Nature Communications Epi of Pm and Po in DRC review

This is a well written article summarizing a basic yet useful epidemiological study of the prevalence and incidence of non-falciparum and falciparum infections over three years in the

DRC. A few critical methodological details should be clarified, including the justification for calculation of period prevalence and the overall sampling.

Methods

1. It would be helpful to provide a bit more detail on the sampling, even though the authors note these are described in further detail elsewhere. For example, how were the villages selected? Was probability sampling used allowing for population-based estimates?

Author Reply: Thank you for this comment. See response to Reviewer 1 regarding selection of the urban neighborhood. We have edited the manuscript to provide more detail on sampling as requested.

All households are located within the clinical catchment areas of the study clinics. For two of the Bu (rural) and Kimpoko (peri-urban) health areas, the study clinics selected for the study are the main government clinics serving each health area. The included villages within those two health areas were selected for the study because of their proximity to the study clinics. These villages were either within the same village as the study clinic or within 5km distance away. For the urban Voix du Peuple health area, more health clinics were available due to the densely populated urban setting. We therefore selected the neighborhood of the catchment of a private study clinic that was used by the most diverse population in that neighborhood in terms of wealth.

Revisions: (Pg. 6), “Three local health facilities were included in our study, one per health area. The chosen facility was the main government clinic in two of these health areas (Bu – rural, Kimpoko – peri-urban). In urban Voix de Peuple, one private clinic was selected due to the diverse patient population it served. Only households within the clinic catchment areas that were located within 5km from the clinic were eligible for the study.”

2. Additionally, were visits at health facilities only from enrolled participants at the baseline household visit included? It appears this is the case but it would be very helpful to clarify these details more explicitly.

Author Reply: This is correct, we only collected samples and data during health facility visits by enrolled participants. In this study, included participants were those who had DBS samples from the baseline household enrollment visit. Clinic visits among participants who did not have a prior baseline visit and DBS sample were not included. We have clarified this in the manuscript.

Revisions: (Pg. 8), “1) The “survey-based” analysis consists of participant survey data and clinical samples collected at active surveillance household visits for all those enrolled at baseline, and 2) the “clinic-based” analysis consists of participant clinical survey data and clinical samples collected from the subset of the survey-based population who had samples collected at the baseline household enrollment visit and who visited study clinics as-needed when symptomatic during follow-up (passive surveillance).

3. Page 9: “Prevalences of *P. malariae*, *P. ovale* spp., and *P. falciparum* were estimated overall, and by infection type, calculated as the number of infections detected out of the total number of visits in the study”. Do you mean to say that “Prevalence of Plasmodium infection was estimated overall, and by species type (*P. malariae*, *P. ovale* spp., and *P. falciparum*)....”?

Author Reply: This is correct. We have revised the text to read “by species.”

Revisions: (Pg. 10), “Prevalences of *P. malariae*, *P. ovale* spp., and *P. falciparum* were estimated overall, and by species ~~infection~~ across follow-up.”

4. Also, this is a rather unusual definition of period prevalence. Period prevalence is typically defined as the number of people with the disease/condition at any time during the period. To me it does not make sense to use the number of visits as the denominator—why not simply the number of people included during the time period and the numerator is any infection during that time period? Please justify the unusual methodology used to calculate period prevalence.

Author Reply: This was initially calculated at the per-visit level to account for repeated sampling across time and the possibility of multiple infection events throughout follow-up. This method assessed the number of infection events, rather than the number of subjects infected regardless of multiple infections experienced. In doing so, this method therefore assumed all infections were acute events, rather than the same ongoing infections detected at multiple screening points. We have removed the terminology of ‘period prevalence’ to avoid confusion in the denominator used. We have also updated the manuscript to report prevalence as suggested following typical convention, as the number of participants ever infected out of the total number of participants in each population. We also kept characterization of the proportion of all tested samples that were positive for an infection across all follow-ups, to distinguish infected participants vs. total number of infections detected during the approximately 3-year follow-up period.

Revisions: (Pg. 10), “Prevalences of *P. malariae*, *P. ovale* spp., and *P. falciparum* were estimated overall, and by species infection across follow-up. ~~calculated as the total number of infections detected out of the total number of visits in the study.~~ Factors associated with non-falciparum infection prevalences were evaluated using crude linear binomial generalized estimating equation (GEE) models accounting for repeated testing of participants over time.”

(Pg. 14-15), “~~Crude prevalence of non-falciparum infection was lower in household survey visits than symptomatic clinic visits, as expected.~~ **Among 1,565 participants across all baseline and follow-up household surveys in the approximately 2-year follow-up period, 159 (10.2%) participants had 186 PCR-confirmed *P. malariae* infections, 70 (4.5%) participants had 78 *P. ovale* spp. infections, and 990 (63.3%) participants had 1,976 *P. falciparum* infections detected. Of all 5,682 tested samples collected from these participants across household follow-up surveys, 3.3% (95% CI: 2.8-3.8%) were PCR-positive for a *P. malariae* infection, 1.4% (95% CI: 1.1-1.7%) for a *P. ovale* spp. infection, and 34.9% (95% CI: 33.2%-36.5%) for a *P. falciparum* infection. Across all household surveys, we observed a crude period prevalence of 3.3% (95% CI: 2.8%-3.8%) for *P. malariae* infection, and 1.4% (95% CI: 1.1%-1.7%) for *P. ovale* spp. infection. Prevalence of *P. falciparum* infection was higher at 34.9% (95% CI: 33.2%-36.5%).**”

(Pg 15-16), “Similar prevalences to those detected in the survey population were observed among the symptomatic clinic subpopulation, ~~for *P. malariae* (4.0% [95% CI: 3.3%-4.7%]) and *P. ovale* spp. (2.8% [95% CI: 2.2%-3.4%]) infection.~~ **despite longer clinic-based follow-up. Among 1,050 participants who visited study clinics in the approximately 3-year clinic-based surveillance follow-up period, a total of 120 (11.4%) patients had 135 *P. malariae* infections detected, and 80 (7.6%) patients had 95 *P. ovale* spp. infections detected. However, prevalence of *P. falciparum* was higher in symptomatic clinic patients compared to the survey population, with 789 (75.1%) patients having 2,009 *P. falciparum* infections. From the clinic subpopulation, 4.0% (95% CI: 3.3-4.7%) of all 3,407 tested clinic samples across the clinic-based surveillance follow-up period were PCR-positive for a *P. malariae* infection, 2.8% (95% CI: 2.2-3.4%) for a *P. ovale* spp. infection, and 58.7% (95% CI: 56.5-60.8%) for a *P. falciparum* infection.**”

5. Also please clarify how the differences in prevalences were calculated. Page 9 of the methods indicates that binomial GEE was used but was one adjusted model used? Or was each risk factor (presented in Figure 5) evaluated on its own? Please add this detail.

Author Reply: Prevalence differences were calculated from linear binomial GEE models, accounting for repeated visits over time by participants. Prevalence difference models were all run individually as unadjusted models only, to report the crude association between each individual factor and the outcome of *P. malariae*, *P. ovale*, or *P. falciparum* infection. We have clarified in the manuscript that these were crude associations only,

modeled individually. We also note in the discussion that there are no assumptions of causality in this study, therefore risk factor results should be interpreted only as associations.

Revisions: (Pg. 10), “Factors associated with non-falciparum infection prevalences were evaluated using crude linear binomial generalized estimating equation (GEE) models accounting for repeated testing of participants over time. Possible factors associated with malaria infection were modeled individually. An exchangeable working correlation matrix was assumed for GEE models, and robust standard errors were estimated for calculation of 95% confidence intervals (CIs).”

In clarifying the methods for modeling these associations, we also added discussion of the rationale for use of GEE models in this analysis rather than a generalized linear mixed model that would account for subject-level clustering within households, and household-level clustering within the 7 villages of the study design. Modeling this nested clustering structure is too complex given the low number of outcome events that were detected in the study for *P. malariae* and *P. ovale* spp., therefore GEE models were used that addressed the repeated nature of testing across follow-up.

Revisions: (Pg. 10), “While three-level clustering of participants nested within households, and households within village sites is present in the study³¹, this clustering structure was too complex to account for using mixed-effect modeling because of the low number of *P. malariae* and *P. ovale* outcomes detected in this study. Therefore, the GEE model was used. As a sensitivity analysis, we compared GEE results for *P. falciparum* outcomes (which occurred frequently in the study) to results from a generalized linear mixed model accounting for random effects of clustering and repeated testing of subjects over time (Supplementary Table 3).”

Discussion

Pages 17-18: “The high proportion of *P. malariae*-infected participants who experienced multiple infections in the study (24%) is notable given that this species is known to recrudescence.” What evidence is there for recrudescence of *P. malariae* infections? Vivax and ovale infections are known to relapse as they have dormant liver stages, but malariae infections do not have hypnozoites. Please elaborate on this statement and provide references for the assertions made.

Author Reply: Thank you for this comment. See response to Reviewer 1 -- this was an error on our part. We have edited the manuscript to properly characterize the ability of *Plasmodium malariae* to persist long-term at low parasite densities in the blood. Citations have also been added.

Revisions:

(Pg. 20), “The high proportion of **participants with *P. malariae*** infected participants who experienced ~~had~~ multiple infections **infection detected more than once during the study** (24%) is notable given that this **parasite** species can **persist in the blood at low parasite density for long periods of time**^{25,42,43}. Chronic *P. malariae* infection may be associated with deleterious clinical outcomes...”

(Pg 20), “Though we could not distinguish whether multiple *P. malariae* infections **detected in an individual** were **due to** chronic occurrences or acute re-infection events...”

Table 2: The title of the table is slightly misleading; it only mentions non-falciparum infection but falciparum infection is also presented. Can you amend to include mention of falciparum infection in the title please?

Author Reply: We have updated the title of this table as suggested.

Revisions: (Pg. 32, Table 2 title), “ **Table 2. Incidence of infection during household survey visits and symptomatic clinic visits, by *Plasmodium* species.**”

Table 2: How is the baseline (and year 1 and year 2) at-risk population defined for the clinic sub-population? Are these participants who were negative for the malaria species at the baseline household visit AND who had at least one clinic visit during the follow-up period? Or is this otherwise defined? Please add more details to footnote 2 for the clinic sub-population.

Author Reply: This is the correct interpretation. We have edited the table footnotes to clarify this at-risk population.

Revisions: (Pg. 32, Table 2 footnote 2), “ ² **The** at-risk population comprises subjects who were PCR-negative for each species-specific infection at Baseline. **For those in the clinic subpopulation, the at-risk population is defined as participants who were PCR- negative for the malaria species at baseline, and had at least one visit to study health clinics during the follow-up period.**”

Table 2 and Figure 3: I am having a hard time understanding how the number of new infections and the at-risk population compares between Table 2 and Figure 3. The numbers of infections do not seem to quite add up to the household plus clinic population. There are no household

events reported during year 2 and the number of new infections during year 2 in panel b of Figure 2 does not equal those reported in Table 2. Please clarify.

Author Reply: We have updated Table 2 to specify cumulative events in order to improve interpretability. The numbers in table 2 and figure 3 will not perfectly sum together since Figure 3 reflects the combined population. In the combined, 'total' population, events which occurred first at a clinic visit would be shown, while that same event would not be included in the 'survey population' column of table 2, but instead would only be captured in the clinic population column. Instead, if that same subject had another infection at a later survey visit, the first survey infection event would be shown in the survey population column of table 2, since these incidence metrics capture time to first infection only.

In the process of reviewing these numbers, we realized that a larger denominator is appropriate for our incidence estimates. Specifically, for the total population we updated the at-risk population to ensure that subjects who never had a clinic visit during the study, would continue to be included in the total at-risk population until either experiencing an event at a survey visit, being censored, or reaching the end of the full study period. As this increased the at-risk population denominator across the study period, the cumulative incidence estimates decreased slightly for *P. malariae*, *P. ovale* spp. and *P. falciparum*. While our incidence estimates in Figure 3 change as a result, our overall findings and discussion are unchanged with this adjustment.

Revisions:

(Pg. 32, Table 2) Updated numbers in table 2.

(Fig 3 and Fig 4 – separate files uploaded) Updated numbers in Figure 3 and Figure 4.

Supplemental Table 3: Are the characteristics included in this table those of the participants at the baseline household visit? And the PCR positive are those with any positive PCR at the clinic at any time during the follow up period? Some clarifying notes would be helpful.

Author Reply: This table is presenting characteristics of participants at the baseline household visit only for those participants who are a part of the clinic subpopulation (i.e., baseline characteristics for the subset of participants who ever visited a study health clinic during the study). The PCR positives in this table represent the proportion of participants in the clinic subpopulation who tested positive for each species of malaria at the baseline household visit. We believe this information is important to present in order to indicate what proportion of this subpopulation was PCR-positive at baseline and therefore dropped from all incidence estimates. We have added footnotes to this table to clarify what is being presented.

PCR results for all clinic subpopulation samples across the 34-month passive surveillance period are summarized in Supplemental Table 6 for all species.

Revisions: (Supplemental Material, Table 3, Pg 5),

: “ 1. The clinic subpopulation comprises a subset of the survey-based population who had at least 1 symptomatic clinic visit during the study period. This table presents baseline characteristics only for the clinic subpopulation, overall, and broken down by whether the participant was PCR positive for each *Plasmodium* species at the baseline visit.

2. Bu health area is classified as rural, Kimpoko health area as peri-urban, and Voix du Peuple as urban.”